# The potential impact of COVID-19 vaccination on patients with immune thrombocytopenic purpura: A protocol for systematic review and meta-analysis

Yangyang Li[1☯], Demin Kong[1☯], Yicheng Ding[1], Jinhuan Wang[2]*

1 Heilongjiang University of Chinese Medicine, Harbin, China, 2 First Affiliated Hospital Heilongjiang University of Chinese Medicine, Harbin, China

☯ These authors contributed equally to this work.
* wjh_0304@163.com

## Abstract

### Background

Immune thrombocytopenic purpura (ITP) is a disease characterized by a decrease in platelet count, which can be triggered by various factors, including viral infections and vaccination. With the widespread vaccination against COVID-19, concerns have arisen regarding a possible link between the vaccine and the exacerbation of ITP. This study aims to comprehensively evaluate the impact of COVID-19 vaccination on ITP patients, including associated risks and outcomes.

### Methods

A comprehensive search will be conducted in multiple electronic databases (including PubMed, Web of Science, EMBASE, Cochrane Library, CNKI, Wan Fang, VIP, and CBM) to identify relevant studies. This study will include randomized controlled trials, cohort studies, case-control studies, and case series evaluating the impact of COVID-19 vaccination on ITP patients. The primary outcome measure is the change in platelet count, while secondary outcome measures include the occurrence of thromboembolic events, bleeding complications, ITP recurrence rate, impact of ITP exacerbation, and adverse events. Data will be summarized and analyzed using Review Manager Software (RevMan) V.5.4. In addition, subgroup analyses will be performed to explore potential sources of heterogeneity.

### Results

It is anticipated that different types of COVID-19 vaccines may have varying impacts on ITP patients, leading to potential differences in outcomes. This study aims to comprehensively evaluate the potential impact of COVID-19 vaccination on ITP patients and provide reference for clinical decision-making.

**Data Availability Statement:** All data supporting the findings of this study are available within the paper and its Supplementary Information.

**Funding:** The author(s) received no specific funding for this work.

## Conclusions

The results of this systematic review and meta-analysis will provide crucial information on COVID-19 vaccination for ITP patients and clinicians, contributing to guiding vaccination decisions and monitoring potential impacts after vaccination.

## Introduction

### Description of the condition

The autoimmune disorder known as immune thrombocytopenic purpura (ITP) is characterized by a decreased platelet count [1,2], which poses a risk of bleeding and has a significant impact on patients' quality of life [3,4]. This condition can be triggered by various inflammatory factors, such as infections or vaccinations [5], and has been further complicated by the global COVID-19 crisis [6–8]. This has raised concerns about the potential adverse effects of COVID-19 vaccination in individuals with ITP [9,10].

Recently, there has been a growing focus on cases or worsening of ITP associated with COVID-19 vaccination [11]. Multiple case reports have documented new occurrences or relapses of ITP in patients who have received COVID-19 vaccination, as well as after being infected with the virus [12–15]. Although there is still ongoing debate about the causal relationship between COVID-19 vaccination and new cases of ITP, there have been reports of 16 ITP episodes following vaccination [16–18]. Therefore, it is strongly recommended to closely monitor individuals after vaccination, considering the observed exacerbation of ITP in some patients. Research has shown that ITP patients experience a significant 6.3% decrease in platelet counts compared to healthy controls after receiving COVID-19 vaccination [19]. While SARS-CoV-2 vaccines are generally regarded as safe for patients with preexisting ITP, they may contribute to worsening thrombocytopenia, particularly for patients who have undergone splenectomy or received multiple therapeutic interventions [20]. Despite confirmed cases of ITP development following COVID-19 vaccination, the exact relationship between vaccination and primary or secondary ITP remains unclear [21].

### Description and function of intervention

The intervention in this systematic review and meta-analysis is COVID-19 vaccination. Vaccination is a widely used public health measure to prevent infectious diseases by inducing the immune system to produce antibodies against a specific pathogen. In the context of COVID-19, vaccination aims to reduce the risk of infection, severe illness, and death by triggering an immune response against the SARS-CoV-2 virus. However, the impact of COVID-19 vaccination on patients with immune-mediated diseases, such as ITP, is not fully understood.

### Why it is important to do this review

The importance of conducting a systematic review and meta-analysis on the potential impact of COVID-19 vaccination on ITP patients lies in several aspects. Firstly, ITP patients are a vulnerable population that may experience adverse effects from vaccination due to their underlying immune disorder. Evaluating the safety and efficacy of COVID-19 vaccines in this population is crucial to ensure their protection against the pandemic. Secondly, the findings of this review can provide valuable insights for clinicians in making informed decisions regarding vaccination for ITP patients. This is especially relevant in the context of the ongoing COVID-19 pandemic, where vaccination remains a key strategy in containing the spread of the virus. Finally, this review can contribute to the growing body of evidence on the safety and efficacy

of COVID-19 vaccines in special patient populations, furthering our understanding of the immune response to vaccination in these groups.

## Methods

### Study registration

The current research project has been duly recorded in the PROSPERO (registration number CRD42023471315) of the International Prospective Register of Systematic Reviews. The research procedure adheres to the guidelines specified in the PRISMA-P (Preferred Reporting Items for Systematic Reviews and Meta-Analysis Protocol) statement [22]. The detailed PRISMA-P checklist is attached in S1 Appendix.

### Inclusion and exclusion criteria

**Type of study.** The assessment of the impact of COVID-19 vaccines on patients with immune thrombocytopenia (ITP) encompasses randomized controlled trials (RCTs), cohort studies, case-control studies, and case series.

**Type of participants.** The eligibility requirements comprise case studies and collections of cases analyzing the association between ITP and COVID-19 infection, as mentioned in the citations [23–26]. To define ITP, the literature will be utilized, specifically excluding any other potential causes or types of thrombocytopenia. Any studies conducted in languages other than English or those addressing different causes or types of thrombocytopenia apart from ITP will not be included in the analysis.

**Type of intervention.** We aim to incorporate all COVID-19 vaccination treatment interventions, encompassing diverse categories, dosages, regimens, and manufacturers. Moreover, we will take into account the administration of COVID-19 vaccination with varying intervals and sequences.Our research will explore the effectiveness of various comparator interventions, including:

1. Individuals who have not been administered the COVID-19 vaccine.

2. Participants who have received a placebo or sham vaccination.

3. Subjects who have undergone other vaccinations or taken different medications.

4. Individuals who have received alternate types of COVID-19 vaccines.

5. Participants who have received varying dosages or treatment schedules for the COVID-19 vaccine.

**Type of outcome measures.** The main focus of the study will be on the alteration in platelet count as the primary outcome.Secondary outcomes that will be evaluated comprise the occurrence of thromboembolic events, complications associated with bleeding, recurrence rate of ITP, the impact of ITP exacerbation, and any adverse events.

**Search strategy.** In order to carry out an extensive investigation, relevant terms and keywords will be utilized to search significant electronic databases like PubMed, Web of Science, EMBASE, Cochrane Library, CNKI, Wan Fang, VIP, and CBM. Moreover, reference lists of pertinent studies and clinical trial registries will be thoroughly examined to identify potential eligible studies [27]. Research papers published in both English and Chinese from the inception of the databases until December 12, 2025, were examined using the specified search terms: "idiopathic thrombocytopenic purpura", "immune thrombocytopenic purpura", "ITP", or "immune thrombocytopenia"; "randomized controlled trial", "controlled trial", "random allocation", "prospective

study", and "clinical trial"; as well as "COVID-19 Vaccines", "Vaccines", "Vaccination", "Vaccin*", and "sars-cov-2 vaccine". For detailed information on the search strategy, please refer to the Table 1. Uniform strategies will be consistently applied to all additional databases.

## Data collection

**Study selection.** The process of screening studies will include rigorous literature screening, data extraction and management, and thorough examination. The entire process will be undertaken by two independent researchers (Yangyang Li and Demin Kong) who will conduct an initial assessment of the titles, abstracts and keywords of all retrieved studies to identify those that meet the predefined inclusion criteria. To ensure a comprehensive assessment, the full text of all potentially relevant studies will be obtained. In case of any discrepancies or disagreements, an in-depth discussion between the two researchers, with the participation of the third author (Jinhuan Wang), will be conducted to resolve any inconsistencies and reach a consensus. The PRISMA flow chart is shown in Fig 1. Fig 1 provides a clear illustration of the comprehensive selection process.

**Assessment of risk of bias in included studies.** The Cochrane Risk of Bias tool will be utilized by reviewers to evaluate 7 domains: random sequence generation, allocation concealment, blinding of participants and personnel, blinding of outcome assessment, assessment of incomplete data, selective outcome reporting, and identification of other sources of bias. The level of uncertainty, low risk, and high risk will be used to assess each domain of bias.

**Dealing with missing data.** If possible, we will strive to establish contact with the authors of the study in order to obtain any missing data or seek clarification. To assess the potential impact of the missing data, the following strategies will be employed:1.A worst-case scenario analysis will be conducted, wherein all participants with missing data will be considered as failures.2.An extreme worst-case analysis will be performed, where participants with missing data in the experimental group will be deemed failures, while those with missing data in the control group will be regarded as successes.3. An extreme best-case analysis will be undertaken, whereby participants with missing data in the experimental group will be considered successes, and those with missing data in the control group will be regarded as failures.

**Table 1. The search strategy for PubMed.**

| ORDER | STRATEGY |
|---|---|
| #1 | Search: "Idiopathic Thrombocytopenic Purpura"[Mesh] |
| #2 | Search: "Thrombocytopenic Purpura, Idiopathic"[Title/Abstract] OR "Immune Thrombocytopenic Purpura" OR "ITP" OR "Immune Thrombocytopenia"[Title/Abstract] |
| #3 | #1 OR #2 |
| #4 | Search: "COVID-19 Vaccines"[Title/Abstract] OR "Vaccines" [Title/Abstract] OR "Vaccination"[Title/Abstract] OR "Vaccin*" [Title/Abstract] OR "sars-cov-2 vaccine" [Title/Abstract] |
| #5 | Search: "randomized controlled trial"[Publication Type] OR "RCT randomized controlled"[Publication Type] OR "random allocation" [Title/Abstract] OR "allocation, random"[Title/Abstract] OR "randomized, controlled"[Title/Abstract] OR "clinical trial" [Title/Abstract] OR "prospective Studies" [Title/Abstract] |
| #6 | Search: "humans"[MeSH Terms] NOT "animals"[MeSH Terms] |
| #7 | #5 AND #6 |
| #8 | #3 AND #4 AND #7 |

ITP, immune thrombocytopenic purpura; COVID-19, corona virus disease 2019; RCT, randomized controlled trial.

**PRISMA 2020 flow diagram for new systematic reviews which included searches of databases and registers only**

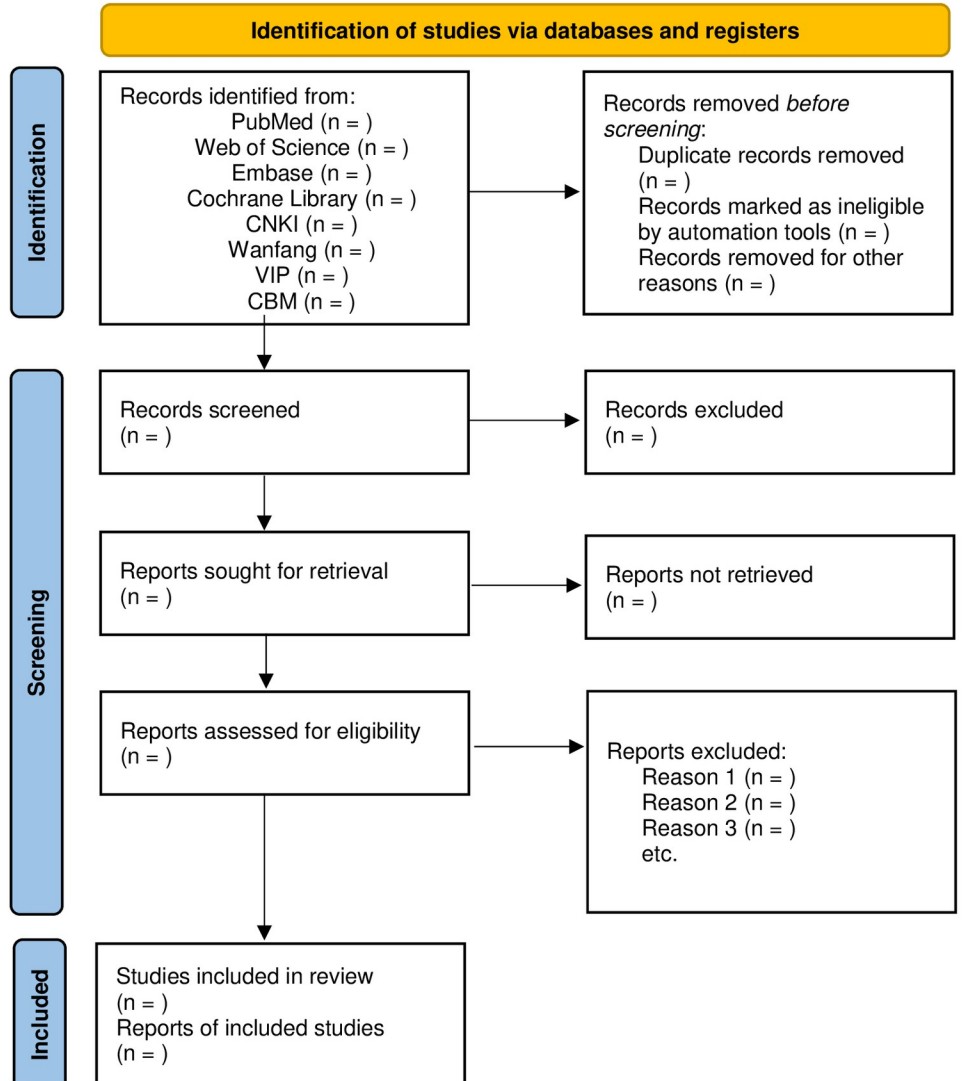

*Consider, if feasible to do so, reporting the number of records identified from each database or register searched (rather than the total number across all databases/registers).

**If automation tools were used, indicate how many records were excluded by a human and how many were excluded by automation tools.

**Fig 1. A PRISMA-P-guided flowchart illustrating the article selection process.** (n): Denotes the cumulative count of articles to be incorporated at each sequential stage. *From*: Page MJ, McKenzie JE, Bossuyt PM, Boutron I, Hoffmann TC, Mulrow CD, et al. The PRISMA 2020 statement: an updated guideline for reporting systematic reviews. BMJ 2021;372:n71. doi: 10.1136/bmj.n71.

## Data synthesis

When examining studies that investigate the same treatment and produce similar outcomes in comparable populations, we will utilize meta-analysis to combine multiple trials and estimate

the overall treatment effect. In order to pool continuous data, we will employ the inverse variance method, while for dichotomous data, we will use the Mantel-Haenszel method. In cases where there is low statistical heterogeneity, we will choose the fixed-effect model for data synthesis. However, if the p-value is below 0.1 or the $I^2$ value exceeds 50%, we will employ the random-effect model to provide a more cautious estimation of the effect. All analyses will be conducted using Review Manager V.5.4 software. If it is not possible to conduct a meta-analysis, we will present a narrative summary of the individual study findings.

### Subgroup analysis

To examine potential heterogeneity sources, such as variations in study designs, patient demographics, and COVID-19 vaccine types administered, subgroup analyses will be performed. These analyses will provide valuable insights into the potential impact of these factors on the outcomes of interest.

### Sensitivity analysis

To assess the robustness of the findings, sensitivity analyses will be carried out, which will involve evaluating the impact of individual studies on the overall results. These analyses aim to provide a deeper comprehension of the stability and reliability of the results obtained from the meta-analysis.

### Publication bias

In the event that our meta-analysis encompasses more than 10 studies, we shall conduct a thorough evaluation of potential publication bias, utilizing the EGGER regression test as our preferred methodology. Following this assessment, we will present the outcomes in a visually compelling manner, employing funnel plots to illustrate any detected biases.

### Grading the quality of evidence

In evaluating the quality of evidence, we will adopt the Grading of Recommendations, Assessment, Development, and Evaluation (GRADE) system as a guiding principle [28]. This system provides a four-level evaluation scale, namely high, moderate, low, and very low quality. The specific classification criteria are as follows: high quality (no evidence of downgrading), moderate quality (one evidence of downgrading), low quality (two evidences of downgrading), and very low quality (more than two evidences of downgrading).Based on the efficacy evaluation standards, we can categorize the overall clinical effectiveness rate into two levels, namely level 3 and level 4. Level 3 includes cure/markedly effective, effective, and ineffective outcomes, while level 4 comprises cure, markedly effective, effective, and ineffective outcomes.

### Ethics and dissemination

This systematic literature analysis and meta-synthesis will uphold ethical principles and standards. Approval from the appropriate ethics review committee will be secured prior to conducting the investigation, ensuring the safeguarding and confidential handling of all individual data involved. Our aim is to disseminate our research findings comprehensively by publishing the final systematic literature analysis and meta-synthesis report upon the completion of the study. We plan to publish the outcomes in a peer-reviewed scientific journal and present them orally at academic gatherings to convey our research findings to the academic community and clinical practitioners. Additionally, we will disseminate the research results to

the public through social media platforms and scientific communication channels in order to enhance awareness regarding the impact of COVID-19 vaccines on patients with ITP.

## Discussion

According to data from the World Health Organization (WHO), as of November 2023, 66% of the global population has been fully vaccinated against COVID-19 with the primary series of vaccines. The effectiveness of these COVID-19 vaccines in preventing severe COVID-19 outcomes has been widely recognized worldwide, playing a crucial role in maintaining public health safety. While these vaccines have demonstrated high immunogenicity, there have been persistent concerns regarding the potential for severe adverse events (AEs) following COVID-19 vaccination [29]. However, for the specific group of patients with immune thrombocytopenic purpura (ITP), the potential risks and benefits associated with vaccination deserve further exploration [30,31].

Due to their low platelet counts, ITP patients face an increased risk of bleeding, rendering the potential immune response following vaccination a significant concern. Although most vaccine clinical trials have not included ITP patients, available data indicate that some ITP patients may experience fluctuations in platelet counts after vaccination, increasing their risk of bleeding [32]. Therefore, understanding and assessing the impact of vaccines on ITP patients is paramount.

The clinical heterogeneity of ITP patients plays a crucial role in vaccination decision-making. Factors such as disease severity, treatment history, and platelet count stability can influence patients' responses to vaccines. Patients undergoing immunosuppressive therapy or with a history of severe thrombocytopenia may face higher risks after vaccination. Hence, doctors must consider individual patient circumstances comprehensively to ensure scientific and reasonable vaccination decisions.

While current evidence is limited, preliminary studies suggest that COVID-19 vaccines may be safe and effective for some ITP patients. However, close monitoring after vaccination is crucial for this special population. Doctors should regularly check patients' platelet counts, assess bleeding symptoms, and be vigilant for other potential adverse events. Through continuous monitoring and evaluation, we can gain a better understanding of the safety and effectiveness of vaccines among ITP patients.

To further explore the impact of COVID-19 vaccines on ITP patients, future research needs to focus on larger-scale and longer-term clinical trials. These studies should investigate the effects of vaccines on different clinical subgroups, such as patients under treatment, in remission, or with unstable platelet counts, and assess the long-term safety and effectiveness of vaccines. Moreover, research should address specific immune responses that may occur after vaccination and their impact on ITP patients. By comprehensively evaluating patients' clinical status, disease characteristics, and vaccine safety data, we can develop more personalized and reasonable vaccination recommendations. With more research and data accumulation in the future, we will gain a deeper understanding of the impact of COVID-19 vaccines on ITP patients, providing more solid evidence support for clinical decision-making.

## Supporting information

**S1 Appendix. The PRISMA-P checklist.**
(DOCX)

## Acknowledgments

The authors would like to thank those who provided comments on the revision of this review.

## Author Contributions

**Conceptualization:** Yangyang Li, Jinhuan Wang.

**Data curation:** Yangyang Li, Demin Kong.

**Formal analysis:** Yangyang Li, Demin Kong, Yicheng Ding.

**Investigation:** Yangyang Li, Demin Kong, Jinhuan Wang.

**Methodology:** Yangyang Li, Demin Kong.

**Project administration:** Yangyang Li, Jinhuan Wang.

**Resources:** Yangyang Li, Demin Kong, Yicheng Ding.

**Software:** Yangyang Li, Demin Kong, Yicheng Ding.

**Supervision:** Jinhuan Wang.

**Validation:** Yangyang Li, Demin Kong, Jinhuan Wang.

**Visualization:** Yangyang Li, Demin Kong, Jinhuan Wang.

**Writing – original draft:** Yangyang Li.

**Writing – review & editing:** Yangyang Li, Demin Kong, Jinhuan Wang.

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
