## [Decision Letter · Decision Letter 0]

26 Jul 2024

The potential impact of COVID-19 vaccination on patients with immune thrombocytopenic purpura：A protocol for systematic review and meta-analysis

PONE-D-24-20892

Dear Dr. Wang,

We’re pleased to inform you that your manuscript has been judged scientifically suitable for publication and will be formally accepted for publication once it meets all outstanding technical requirements.

Kind regards,

Mehmet Baysal

Academic Editor

PLOS ONE

file:///home/jasmine/Downloads/journal.pone.0300769.pdf

In your revision ensure you cite all your sources (including your own works), and quote or rephrase any duplicated text outside the methods section. Further consideration is dependent on these concerns being addressed.

3. PLOS requires an ORCID iD for the corresponding author in Editorial Manager on papers submitted after December 6th, 2016. Please ensure that you have an ORCID iD and that it is validated in Editorial Manager. To do this, go to ‘Update my Information’ (in the upper left-hand corner of the main menu), and click on the Fetch/Validate link next to the ORCID field. This will take you to the ORCID site and allow you to create a new iD or authenticate a pre-existing iD in Editorial Manager. Please see the following video for instructions on linking an ORCID iD to your Editorial Manager account: https://www.youtube.com/watch?v=_xcclfuvtxQ".

Reviewers' comments:

Reviewer's Responses to Questions

**Comments to the Author**

1. Does the manuscript provide a valid rationale for the proposed study, with clearly identified and justified research questions?

Reviewer #1: Yes

Reviewer #2: Yes

2. Is the protocol technically sound and planned in a manner that will lead to a meaningful outcome and allow testing the stated hypotheses?

Reviewer #1: Yes

Reviewer #2: Yes

3. Is the methodology feasible and described in sufficient detail to allow the work to be replicable?

Reviewer #1: Yes

Reviewer #2: Yes

4. Have the authors described where all data underlying the findings will be made available when the study is complete?

Reviewer #1: Yes

Reviewer #2: Yes

5. Is the manuscript presented in an intelligible fashion and written in standard English?

Reviewer #1: Yes

Reviewer #2: Yes

6. Review Comments to the Author

You may also provide optional suggestions and comments to authors that they might find helpful in planning their study.

Reviewer #1: The study is registered in PROSPERO, which indicates transparency and adherence to systematic review protocols. Additionally, PRISMA-P guidelines are followed, ensuring standardized reporting. The authors describe that they will include randomized controlled trials, cohort studies, case-control studies and case series, ensuring a broad scope of evidence.

Various vaccine schedules against COVID-19 are considered, including different manufacturers, dosages and schedules.

A comprehensive search strategy will be used using multiple and relevant databases (PubMed, Web of Science, etc.) as well as specific search terms related to ITP, COVID-19 vaccines, and study types. A rigorous screening process involving two independent reviewers will be used to select studies and extract data. Additionally, resolution of discrepancies through consensus will be done by a third reviewer. The Cochrane Risk of Bias tool will be used to evaluate included studies across multiple domains. Strategies for dealing with missing data are predefined. The GRADE system will be adopted to assess the quality of evidence, which provides transparency in assessing confidence in study results. The Approval from the appropriate ethics review committee will be secured prior to conducting the investigation, guaranteeing data confidentiality.

Reviewer #2: The protocolo for the systemic review and meta-analysis seems to be complete. It will be useful for this very specific population, potentially practice changing as it can provide important information to guide clinicians regarding vaccination for COVID 19 in patients with ITP.

7. PLOS authors have the option to publish the peer review history of their article (what does this mean?). If published, this will include your full peer review and any attached files.

Reviewer #1: **Yes: **Gabrielle de Mello Santos

Reviewer #2: No

---

## [Editor Report · Acceptance letter]

31 Oct 2024

PONE-D-24-20892 

PLOS ONE

Dear Dr. Wang, 

I'm pleased to inform you that your manuscript has been deemed suitable for publication in PLOS ONE. Congratulations! Your manuscript is now being handed over to our production team.

Kind regards, 

on behalf of

Dr. Mehmet Baysal 

Academic Editor

PLOS ONE